# The Role of ACE, ACE2, and AGTR2 Polymorphisms in COVID-19 Severity and the Presence of COVID-19-Related Retinopathy

**DOI:** 10.3390/genes13071111

**Published:** 2022-06-21

**Authors:** Kristina Jevnikar, Luka Lapajne, Daniel Petrovič, Andrej Meglič, Mateja Logar, Nataša Vidovič Valentinčič, Mojca Globočnik Petrovič, Ines Cilenšek, Polona Jaki Mekjavić

**Affiliations:** 1Department of Ophthalmology, University Medical Centre Ljubljana, 1000 Ljubljana, Slovenia; kristina.jevnikar@kclj.si (K.J.); luka.lapajne@kclj.si (L.L.); andrej.meglic@kclj.si (A.M.); natasa.vidovic@kclj.si (N.V.V.); mojca.globocnik@kclj.si (M.G.P.); 2Department of Ophthalmology, Faculty of Medicine, University of Ljubljana, 1000 Ljubljana, Slovenia; 3Laboratory for Histology and Genetics of Atherosclerosis and Microvascular Diseases, Institute of Histology and Embryology, Faculty of Medicine, University of Ljubljana, 1000 Ljubljana, Slovenia; daniel.petrovic@mf.uni-lj.si (D.P.); ines.cilensek@mf.uni-lj.si (I.C.); 4Department of Infectious Diseases, University Medical Centre Ljubljana, 1000 Ljubljana, Slovenia; mateja.logar@kclj.si; 5Department of Infectious Diseases, Faculty of Medicine, University of Ljubljana, 1000 Ljubljana, Slovenia

**Keywords:** COVID-19, angiotensin-converting enzyme, angiotensin-converting enzyme 2, angiotensin II receptor type 2, polymorphism, retina

## Abstract

The proposed SARS-CoV-2-induced dysregulation of the renin-angiotensin-aldosterone (RAAS) system results in endothelial dysfunction and microvascular thrombosis. The retinal plexuses contain terminal vessels without anastomotic connections, making the retina especially susceptible to ischemia. This study aimed to determine the role of selected polymorphisms of genes in the RAAS pathway in COVID-19 severity and their association with the presence of COVID-19 retinopathy. 69 hospitalized patients in the acute phase of COVID-19 without known systemic comorbidities and 96 healthy controls were enrolled in this prospective cross-sectional study. The retina was assessed with fundus photography using a Topcon DRI OCT Triton (Topcon Corp., Tokyo, Japan) in the COVID-19 unit. Genotyping of selected polymorphisms in the genes for ACE (rs4646994), ACE2 (rs2285666), and AGTR2 (rs1403543) was performed. The COVID-19 group was divided into mild (*n* = 12) and severe (*n* = 57), and then further divided according to the presence of COVID-19 retinopathy (Yes, *n* = 50; No, *n* = 19). The presence of the AGTR2 rs1403543-AA genotype was associated with a 3.8-fold increased risk of COVID-19 retinopathy (*p* = 0.05). The genotype frequencies of selected gene polymorphisms were not significantly associated with either the presence of COVID-19 or its severity. This is the first study demonstrating a borderline association of the AGTR2 rs1403543-AA genotype with COVID-19 retinopathy in males; hence, the AGTR2 rs 1403543 A allele might represent a genetic risk factor for COVID-19 retinopathy in males.

## 1. Introduction

Coronavirus disease 2019 (COVID-19), caused by severe acute respiratory syndrome coronavirus 2 (SARS-CoV-2), predominantly affects the respiratory system in the form of viral pneumonia; however, several organ systems can be affected [1,2]. SARS-CoV2 enters the cell by binding to angiotensin-converting enzyme 2 (ACE2), expressed on various tissues’ host cell surfaces, including lungs, kidneys, heart, and blood vessels [3,4]. In the retina, it is expressed in the vascular endothelium, Müller glia and ganglion cells, and neurons in the inner nuclear layer [5]. By binding to ACE2, SARS-CoV2 creates an imbalance in the signaling effects of the renin-angiotensin-aldosterone (RAAS) pathway. Angiotensin-converting enzyme (ACE) and ACE2 are two of the key enzymes of the RAAS pathway. ACE catalyzes angiotensin-I (Ang-I) to angiotensin II (Ang-II), then hydrolyzed to Ang 1–7 by ACE2. By binding to ACE2, SARS-CoV2 downregulates its activity and creates an imbalance in the signaling effects of Angiotensin II (Ang II) and its receptor (angiotensin II type 1 receptor, AT1), resulting in the accumulation of Ang II. Increased serum Ang II leads to vasoconstriction, inflammation, cellular differentiation and growth, endothelial dysfunction, the formation of reactive oxidative species, and microvascular thrombosis. On the contrary, Ang II actions mediated through angiotensin II type 2 receptor (AT2) have a vasoprotective and anti-inflammatory role [6,7,8]. Nevertheless, AT2 has been proposed as an alternative entry point of SARS-CoV2, blocking its possible protective role [9]. Hence, an imbalance in the signaling effects of the RAAS pathway results in the hypercoagulable state, predisposing patients to thromboembolic events [8,10]. COVID-19-related retinopathy, defined as the presence of flame-shaped hemorrhages, cotton wool spots, dilated veins, and tortuous vessels, has been reported [10,11]. Notably, the findings were more pronounced in patients with a severe course [10,11]. The retina is especially susceptible to ischemia because of its metabolic demands and the fact that its retinal plexuses contain terminal vessels without anastomotic connections [12,13]. Therefore, its assessment with fundus photography, a non-invasive imaging method, provides a valuable insight into COVID-19 at a microvascular level [14]. COVID-19 has a broad clinical spectrum ranging from asymptomatic to severe. Several factors were shown to affect the clinical course, namely age, male gender, and pre-existing comorbidities, especially cardiovascular disease, diabetes, and hypertension [6,15,16]. It has been hypothesized that the polymorphisms in the genes of the RAAS pathway could also play a role in disease susceptibility and severity [17,18,19,20]. The ACE2 gene is located on the short arm of the X chromosome (Xp22.2) and consists of 22 exons [20]. One of its most studied single-nucleotide polymorphisms (SNP) is rs2285666 at the fourth base of the third intron (G > A). It has been suggested that its locus could alter alternative splicing of messenger RNA (mRNA) and affect ACE2 receptor gene expression [17,21]. Moreover, strong linkage disequilibrium with the other ACE2 receptor SNPs (rs1978124, rs714205) has been found [17]. It has been proposed that its location on the X chromosome could explain the increased prevalence of severe COVID-19 in males [22]. The ACE gene is located on the long arm of human chromosome 17 and consists of 26 exons and 25 introns (17q23.3). The insertion/deletion polymorphism rs464994 is a functional polymorphism present in intron 16 in the form of either insertion (I allele) and/or deletion (D allele) of the 289 bl Alu repeat sequence, which affects the expression of ACE [17,18]. The DD genotype was found to be associated with increased levels of the serum ACE [17,18,19]. AT2 is expressed in several organs’ vascular endothelium, including the retina [23]. While its expression is low in normal conditions, it is upregulated in pathological states [23,24]. The AT2 gene AGTR2 is located on the X chromosome (Xq23). It consists of three exons and two introns. Its SNP rs1403543 is located on intron one (G > A) and has not yet been studied in association with COVID-19; however, the A allele was shown to be associated with cardiovascular disease [24]. In addition, it was shown to affect the retinal arteriole diameter [23]. It is important to note that ACE, ACE2, and AGTR2 polymorphisms have also been associated with hypertension, diabetes, coronary artery disease, and stroke [17,18,19,20,21,22]. As the previous studies did not exclude patients with comorbidities, it remains unclear whether the polymorphisms really play a role in COVID-19 severity or just reflect the higher prevalence of comorbidities in critically ill patients. This study aimed to determine the role of selected polymorphisms in ACE (rs4646994), ACE2 (rs2285666), and AGTR2 (rs1403543) in the presence of COVID-19, its severity, and their association with the presence of the COVID-19 retinopathy using fundus photography in patients without known comorbidities and compare it with healthy, age-matched controls.

## 2. Materials and Methods

### 2.1. Study Design

A prospective cross-sectional study was conducted at the University medical center Ljubljana (UMCL) between December 2020 and December 2021. The study was approved by the Slovenian medical ethics committee (protocol ID number: 0120-553/2020/3) and adhered to the tenets of the Declaration of Helsinki. Written informed consent was obtained from all participants enrolled in the study. 

### 2.2. Patient Selection, Inclusion, and Exclusion Criteria

We included 113 consecutive patients aged 18–65 with PCR-confirmed SARS-CoV-2 admitted to the COVID-19 unit of the department of infectious diseases, UMCL. Sixty-nine participants were included in the final analysis after applying the following exclusion criteria: systemic comorbidities (diabetes, arterial hypertension, hyperlipidemia, coronary artery disease, history of stroke); concomitant infectious diseases (HIV, HSV, VZV, CMV); systemic treatment linked to retinal toxicity, smoking, pre-existing ocular pathology, age-related macular degeneration, or other retinal diseases; and a history of glaucoma, high myopia (>−6) or other conditions that could have affected the retinal morphology. Ninety-six volunteers without comorbidities and no history of COVID-19 represented the age-matched control group. All enrolled participants were Caucasians.

### 2.3. Study Protocol

Electronic medical records of hospitalized patients were reviewed to obtain relevant demographics and clinical characteristics. The following data were collected: age, sex, time from the symptoms’ onset or positive PCR to the day of fundus imaging, the presence of COVID-19-related symptoms, the need for oxygen, COVID-19-related treatment, and outcome. Laboratory parameters included lactate dehydrogenase (LDH), ferritin, C-reactive protein (CRP), procalcitonin, white blood cells, red cell distribution width (RDW), platelets, lymphocytes, D-dimer, and 25-OH-D3. Patients were divided into four groups based on the COVID-19 disease severity classification: (1) asymptomatic (positive PCR, no symptoms), (2) mild (presence of symptoms but no shortness of breath, dyspnoea, or abnormal chest imaging), (3) moderate (evidence of lower respiratory disease during clinical assessment or imaging and an oxygen saturation (SpO2) ≥ 94% on room air at sea level), and (4) severe disease (SpO2 < 94% on room air at sea level, a ratio of arterial partial pressure of oxygen to fraction of inspired oxygen (PaO2/FiO2) < 300 mm Hg, respiratory frequency > 30 breaths/min, or lung infiltrates > 50%) [25]. To facilitate statistical analysis, groups were combined: asymptomatic and mild disease were classified as mild; moderate and severe disease were classified as severe. Peripheral blood samples were collected for genotyping. Fundus images were obtained after dilating the pupils (1% tropicamide) using swept-source OCT (SS-OCT, Topcon DRI OCT Triton; Topcon Corp., Tokyo, Japan). The study protocol consisted of 2 color fundus images per eye (one cantered on the fovea, one on the optic disc). All the images were obtained by two doctors (KJ, LL). The appropriate full-body protective gown with the FPP 3 mask was worn in the COVID-19 unit. 

### 2.4. Image Analysis

Only images with a signal strength index above 60 were included in the analysis. The eyes were excluded from analysis if the fundus details were not visible due to media opacities or acquisition artifacts. Three researchers (KJ, AM, and PJM) independently reviewed the fundus photographs for the presence of retinal findings, namely: retinal hemorrhages, cotton wool spots, or dilated and tortuous vessels. Patients exhibiting any of the aforementioned findings were classified as having COVID-19 retinopathy. Vessel diameters were assessed by processing fundus photographs with the Automated Retinal Image Analyser (ARIA, V1-09-12-11) and MATLAB (MathWorks Inc., Natick, MA, USA), a previously described method [26]. The vessel diameters of the four main veins and four main arteries between 0.5 and 1 disc diameter from the optic disc margin were used to calculate the mean vein diameter (MVD) and mean artery diameter (MAD). 

### 2.5. Genotyping 

DNA was isolated from peripheral venous blood using the QIAamp DNA Mini Kit (Qiagen GmbH, Hilden, Germany). The genotyping of the selected polymorphisms in ACE (rs4646994), ACE2 (rs2285666), and AGTR2 (rs1403543) were performed using KASPar genotyping chemistry with validated assays. Details of the method used can be found on http://www.kbioscience.co.uk/ (accessed on 16 July 2021). 

### 2.6. Statistical Analysis

Numerical variables are reported as median (interquartile range, IQR), and the differences between groups were tested using Mann–Whitney or Kruskal–Wallis tests, as appropriate (normality was checked with Shapiro–Wilk’s tests). Descriptive variables are reported as numbers (%) and compared with the Chi-square test or Fisher’s exact test. The *p* values were adjusted with the Benjamini–Hochberg method. The χ2 test was used to calculate Hardy–Weinberg equilibrium (HWE) and compare the selected polymorphisms’ genotype frequencies. Because the ACE2 and AGTR2 genes are located on the X chromosome, females and males were separately analyzed to account for the fact that males are hemizygous. HWE was calculated using the HardyWeinberg package of the R software. Firth’s logistic regression analysis was used to assess the association between COVID-19 retinopathy and polymorphisms, adjusted for severity and HbA1c. The association between vessel diameters (measured at four sides) and polymorphisms was assessed by a linear mixed-effect model with subject as a random effect and polymorphisms as fixed effects. In the first model, ACE was a fixed effect, and the analysis was adjusted for severity, sex, age, oxygen, LDH, and ferritin. The other two models were fitted separately for females and males, with ACE2 and AGTR2 as fixed effects and adjusted for age, oxygen, LDH, and ferritin. Statistical analysis was performed using R 4.1.3 (R Foundation for Statistical Computing, Vienna, Austria).

## 3. Results

### 3.1. Demographics and Clinical Characteristics

A total of 69 consecutive patients in the acute phase of COVID-19 without known systemic comorbidities and 96 healthy controls were included in the final analysis. Demographics and clinical characteristics of study participants are presented in Table 1. The median age of the COVID-19 group was 58 years (IQR 50–62), and 45 (65%) were male. The median age in the control group was 53 (IQR 41–60). The groups were age-matched (*p* = 0.06). The COVID-19 group was divided according to the disease severity into mild (*n* = 12) and severe (*n* = 57). Cough (*p* < 0.001) and dyspnoea (*p* < 0.001) were significantly more common in the severe course. Patients with a severe course exhibited higher activity of LDH (*p* < 0.001) and higher concentrations of ferritin (*p* = 0.001). The median duration of symptoms before imaging was 3.5 days (IQR: 1–4.5) in patients with mild disease and 10 days (IQR: 8–13) in patients with severe disease.

### 3.2. COVID-19 Retinopathy

The COVID-19 patients were further divided into two subgroups according to the presence of COVID-19 retinopathy during hospitalization (COVID-19 retinopathy, *n* = 50) or absence of any retinal findings (No retinopathy, *n* = 19). The baseline characteristics of patients, based on the presence of COVID-19 retinopathy, are presented in Table 2. Signs of COVID-19 retinopathy are presented in Figure 1. None of the COVID-19 patients reported ocular symptoms such as itching, photophobia, foreign body, conjunctivitis, decreased visual acuity, or vision loss. There were no differences between the two groups regarding age (*p* = 0.31) or sex (*p* = 0.83); however, CRP and the presence of leukocytopenia and lymphocytopenia were significantly higher in the retinopathy group (*p* = 0.001; *p* = 0.01; *p* = 0.03, respectively). In addition, no statistical difference was found between COVID-19-specific treatments, including dexamethasone (*p* = 0.64), remdesivir (*p* = 0.08), and oxygen (*p* = 0.29), and the presence of COVID-19 retinopathy. 

### 3.3. Genotyping

The distribution of genotypes for ACE I/D, ACE2 rs2285666, and AGTR2 rs1403543 in the COVID-19 patients compared with healthy controls and based on the COVID-19 severity is shown in Table 3. There were no deviations from HWE in any of the studied polymorphisms (Table 3). The genotype frequencies of the ACE I/D gene polymorphism were not significantly associated with either the presence of COVID-19 (*p* = 1.00) or its severity (*p* = 0.58). ACE 2 and AGTR2 are present on the X-chromosome; therefore, the data were analyzed separately for each sex. No association was found between the distribution of genotype or allele frequencies of their selected polymorphisms (rs2285666, rs1403543) and the presence of COVID-19 or its severity. We also evaluated the association of selected polymorphisms and the presence of COVID-19 retinopathy. Table 3 presents the distribution of genotypes for ACE I/D, ACE2 rs2285666, and AGTR2 rs1403543 based on the presence of COVID-19 retinopathy. There were no significant differences between the two groups in the genotype distributions of the rs4646994 and rs2285666 polymorphisms. 

In contrast, the rs1403543 polymorphism of the AGTR2 gene demonstrated a borderline difference (*p* = 0.05) in the genotype distribution in males. We used logistic regression analysis to assess whether rs1403543 was independently associated with the presence of COVID-19 retinopathy. After adjusting for severity and HbA1c, using the GG genotype as a reference, the OR for the AA genotype of the rs1403543 polymorphism was 3.8 (95% CI = 0.99–17.7; *p* = 0.05). Nevertheless, no significant association was found between genotype frequencies of the ACE I/D and ACE2 rs2285666 polymorphisms and the presence of COVID-19 retinopathy. The association of mean vein diameters (MVD) and mean artery diameters (MAD) with the selected polymorphisms was also evaluated; however, no significant differences were found (Table 4).

## 4. Discussion

In the present study, we evaluated the association of candidate gene polymorphisms and the presence of COVID-19 and its severity, as well as their role in the presence of COVID-19 retinopathy. To fully elucidate the effect of COVID-19, we included only patients without known comorbidities. The key finding of this study is a borderline association of the AGTR2 rs1403543 AA genotype with a 3.8-fold increased risk of COVID-19 retinopathy in males. There was no association between the selected polymorphisms in ACE and ACE2 and the presence of COVID-19 retinopathy. In addition, we found no association between selected polymorphisms in ACE (rs4646994), ACE2 (rs2285666), and AGTR2 (rs1403543) and COVID-19 severity. Since the pandemic outbreak, the role of RAAS in the pathogenesis of COVID-19 has been widely suspected. Several studies addressed the role of various polymorphisms in the susceptibility and severity of COVID-19; however, the results have been inconclusive. It has been hypothesized that increased gene expression of ACE2, resulting in higher ACE2 serum levels, could affect COVID-19 susceptibility and severity by increasing the number of viral binding sites [20,22]. Even though the A allele was shown to be associated with higher serum levels of ACE2, the GG genotype and G allele carriers were shown to have an increased risk of SARS-CoV2 infection and a severe course of COVID-19 [20]. Thus, a protective role of A allele-related increase in ACE2 has been suggested, as it could help counterbalance the effects of increased Ag II resulting from RAAS dysregulation [20]. Nonetheless, in line with our results, several studies found no association with disease susceptibility or severity [17,21,22]. ACE insertion/deletion polymorphisms (rs4646994 and rs179752) have also been linked to COVID-19 severity. Individuals with a D/D genotype were shown to have a higher incidence of pulmonary embolism and increased mortality rates, possibly due to increased levels of serum ACE [19,27,28]. This differs from our outcomes, otherwise consistent with other studies that found no association between the I/D polymorphism and disease severity [17,20,21]. One of the AGTR2 polymorphisms (rs1914711) was shown to increase the risk of severe COVID-19 in the Mayan population [29]. Nevertheless, our results suggest no association between rs1403543 and disease severity. Several factors could explain the lack of association between the selected polymorphisms and COVID-19 severity. First, the same polymorphisms were shown to affect diabetes, cardiovascular disease, and hypertension [17,18,19,20,21,22]. The studies did not exclude patients with said comorbidities, so their findings could reflect an association with diabetes and hypertension rather than COVID-19 [17,18,19,20,21,22]. Notably, a study by Gómez et al., found that the association of the I/D ACE polymorphism with disease severity depended on the hypertensive status [22]. Moreover, epigenetic regulation of gene expression by post-translational changes such as DNA methylation, histone modifications, and nucleosome positioning could affect COVID-19 susceptibility and severity [17,30]. In addition, pyroptosis, an inflammatory type of programmed host cell death, could also play a role. It leads to the overproduction of inflammatory cytokines and chemokines such as tumor necrosis factor-α, IL-1β, and IL-6, which were shown to trigger the cytokine storm and multiorgan failure [1,2,31]. Hence, genes encoding the said inflammatory cytokines and chemokines could affect the severity of the COVID-19 [1,2,17,31]. To our knowledge, this study is the first to explore the association between the polymorphisms and the COVID-19-related retinal phenotype. Retinal microcirculation comprises terminal vessels without anastomotic connections, making the retina particularly susceptible to hypoxic and ischemic damage resulting from SARS-CoV-2-induced dysregulation of the RAAS system [10]. While no association was found between the selected polymorphisms in the ACE and ACE2 genes, the AGTR2 rs1403543-A allele was associated with a 3.8-fold increased risk of COVID-19 retinopathy in males. This could be ascribed to polymorphism-related differences in AGTR2 expression [23,24]. Warenecke et al. have shown a higher expression of AGTR2 in rs1403543-G allele carriers [24]. It has been suggested that the stimulation of AT2 counteracts the actions mediated through AT1 by three pathways, namely by competing for Ang II, downregulating AT1 expression, and activating the protein phosphatases that downregulate AT1-induced protein kinase activities. In addition, it has been speculated that the stimulation of vascular AT2 receptors increases bradykinin and NO release, which induces vasodilation. Hence, the higher expression reported in G allele carriers could have a protective role. This is in accordance with our results, where a higher risk of retinopathy was associated with the presence of the A allele. The hypothesis of the protective role of the G allele is further supported by the previous studies that have shown an association between the A allele and increased risk of cardiovascular disease—namely, ventricular hypertrophy in men and coronary ischemia in women [24]. One possible explanation for the gender differences also found in our study is that the AGTR2 gene is located on the X-chromosome, making males hemizygous, which affects gene expression. However, the male gender has been established as a risk factor for a severe course of COVID-19, possibly reflecting the gender-based differences in immune responses to COVID-19 [32]. Males were shown to have increased levels of pro-inflammatory chemokines and cytokines, whereas a more robust T-cell response was reported in females [32]. Therefore, our results could reflect the gender-based differences in the immunological response to SARS-CoV2 infection. Another possible explanation for our results is that they reflect a relatively small sample of females, as the *p*-value reaches significance when increasing the sample size. The importance of our findings lies in the fact that retinal vessels reflect the state of the systemic microcirculation [24]. Furthermore, AT2 is expressed in several tissues’ vascular endothelium; hence, similar COVID-19-induced microvascular alterations are likely [24]. The results of this study must be interpreted with caution due to some limitations. First, the sample size of COVID-19 patients is relatively small, resulting from the fact that only patients without known comorbidities were included. Second, the study was limited to the patients admitted to the COVID-19 unit in the study period; therefore, the mild and moderate groups are not size-balanced. Third, our study population was limited to Caucasians; thus, the findings cannot be extrapolated to other populations due to potential ethnic heterogeneity.

## 5. Conclusions

In conclusion, the results of our study suggest a borderline association of the AGTR2 rs1403543 AA genotype with a 3.8-fold increased risk of COVID-19 retinopathy in males. Hence the AGTR2 rs1403543 AA genotype might represent a genetic risk factor for COVID-19 retinopathy in males. However, our findings must be interpreted with caution due to the rather small sample size. In addition, we found no association between selected SNPs in ACE and ACE2 and the presence of COVID-19 retinopathy. Furthermore, no statistically significant association was found between the presence of selected polymorphisms in ACE (rs4646994), ACE2 (rs2285666), and AGTR2 (rs1403543) and COVID-19 severity. Further studies on larger populations are warranted.

## Figures and Tables

**Figure 1 genes-13-01111-f001:**
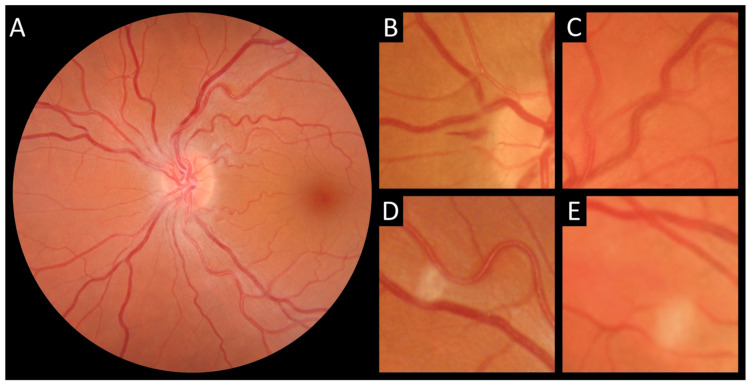
Signs of COVID-19 retinopathy. (**A**) Dilated and tortuous retinal vessels. (**B**) Retinal hemorrhage. (**C**) Dilated and tortuous vein. (**D**,**E**) Cotton wool spots.

**Table 1 genes-13-01111-t001:** Demographic and clinical characteristics of patients in the acute phase of COVID-19, compared with the control group.

	COVID-19		*p*	Controls (*n* = 96)	*p*
	Mild (*n* = 12)	Severe (*n* = 57)			
Age, years	56 (49–60)	58 (50–62)	0.49	53 (41–60)	0.06
Sex (male)	7 (58%)	38 (67%)	0.58	18 (19%)	<0.001
Systolic pressure	125 (114–153)	120 (108–132)	0.14	115 (105, 123)	<0.001
Diastolic pressure	76 (68–83)	72 (69–81)	0.76	79 (75–84)	0.01
**COVID-19 clinical characteristics**					
Duration of symptoms, days	3.5 (1–4.5)	10 (8–13)	<0.001	-	-
Chest pain	1 (8%)	18 (32%)	0.07	-	-
Cough	1 (8%)	55 (96%)	<0.001	-	-
Anosmia/ageusia	0 (0%)	6 (11%)	0.12	-	-
Dyspnoea	2 (17%)	42 (74%)	<0.001	-	-
Diarrhea	0 (0%)	8 (14%)	0.07	-	-
Headache	1 (8%)	10 (18%)	0.40	-	-
DVT/PE	0 (0%)	3 (5%)	0.28	-	-
ICU admission	0 (0%)	6 (11%)	0.12	-	-
**Laboratory parameters**					
HbA1c	5.5 (5.2–5.8)	5.6 (5.4–5.9)	0.18	5.4 (5.1–5.7)	<0.001
LDH (μkat/L)	3.2 (2.4–4.1)	5.0 (4.2–5.8)	<0.001	2.8 (2.5–3.3)	<0.001
Ferritin (μg/L)	196 (110–375)	843 (456–1501)	0.001	53 (27–98)	<0.001
CRP (mg/L)	21 (11–45)	34 (21–57)	0.17	?	
White blood cells (×10^9^/L)	7.7 (5.3–8.5)	6.9 (4.7–8.5)	0.43	7 (5–8.1)	0.58
RDW (%)	14.3 (14.0–15.1)	13.9 (13.3–14.3)	0.06	13 (12–14)	<0.001
Platelets (×10^9^/L)	200 (175–278)	263 (188–331)	0.23	247 (194–317)	0.87
Lymphocytes (×10^9^/L)	1.8 (1.1–1.9)	1.2 (0.8–1.7)	0.09	2.1 (1.7–2.9)	<0.001
D-dimer (μg/L)	742 (427–1117)	711 (513–1388)	0.68	86 (58–251)	<0.001
**Treatment**					
Dexamethasone	0 (0%)	50 (88%)	<0.001	-	-
Remdesivir	0 (0%)	25 (44%)	0.003	-	-
Antibiotic	1 (8%)	8 (14%)	1.00	-	-
Oxygen	0 (0%)	50 (88%)	<0.001	-	-

DVT, deep vein thrombosis; PE, pulmonary embolism; LDH, lactate dehydrogenase; CRP, C-reactive protein; RDW, red cell distribution width.

**Table 2 genes-13-01111-t002:** Baseline characteristics based on the presence of COVID-19 retinopathy.

	COVID-19 Retinopathy		*p*
	NO (*n* = 19)	YES (*n* = 50)	
Age, years	57 (47–60)	58 (50–63)	0.31
Sex (male)	12 (63%)	33 (66%)	0.83
Systolic pressure	120 (111–130)	121 (108–135)	0.86
Diastolic pressure	74 (68–81)	72 (67–83)	0.93
**COVID-19 clinical characteristics**			
Duration of symptoms, days	11.0 (6.5–15.0)	9.0 (6.0–12.8)	0.28
Chest pain	3 (16%)	16 (32%)	0.18
Cough	14 (74%)	42 (84%)	0.33
Anosmia/ageusia	1 (5%)	5 (10%)	1.00
Dyspnoea	11 (58%)	33 (66%)	0.53
Diarrhea	2 (11%)	6 (12%)	1.00
Headache	3 (16%)	8 (16%)	0.98
DVT/PE	0 (0%)	3 (6%)	0.56
ICU admission	3 (16%)	3 (6%)	0.22
**Laboratory parameters**			
HbA1c	5.50 (5.25–5.85)	5.60 (5.40–5.90)	0.31
LDH (μkat/L)	4.40 (3.08–5.18)	4.80 (3.83–5.29)	0.35
Ferritin (μg/L)	816 (310–1434)	684 (374–1447)	0.65
CRP (mg/L)	15 (5–27)	38 (25–64)	<0.001
White blood cells (×10^9^/L)	8.10 (7.20–10.30)	6.60 (4.48–7.97)	0.01
RDW (%)	13.90 (13.45–14.20)	14.05 (13.55–14.55)	0.64
Platelets (×10^9^/L)	289 (215–314)	228 (165–347)	0.30
Lymphocytes (×10^9^/L)	1.67 (1.07–2.03)	1.15 (0.78–1.46)	0.03
D-dimer (μg/L)	590 (474–932)	746 (530–1438)	0.21
**Treatment**			
Dexamethasone	13 (68%)	37 (74%)	0.64
Remdesivir	10 (53%)	15 (30%)	0.08
Antibiotic	3 (16%)	6 (12%)	0.70
Oxygen	12 (63%)	38 (76%)	0.29

DVT, deep vein thrombosis; PE, pulmonary embolism; LDH, lactate dehydrogenase; CRP, C-reactive protein; RDW, red cell distribution width.

**Table 3 genes-13-01111-t003:** The distribution of genotypes for ACE I/D, ACE2 rs2285666, and AGTR2 rs1403543 in the COVID-19 patients compared with healthy controls and based on the presence of COVID-19 retinopathy.

Polymorphism	Genotype	*n*	COVID-19 (*n* = 69)	Controls (*n* = 96)	*p*	*p adj.* ^a^	*n*	COVID-19 Retinopathy		*p*	*p adj.* ^a^
								NO (*n* = 19)	YES (*n* = 50)		
**ACE (rs4646994)**											
Female and male	II	163	11 (16%)	20 (21%)	0.54	1.00	67	3 (17%)	8 (16%)	0.71	0.89
	ID		32 (48%)	49 (51%)				10 (56%)	22 (45%)		
	DD		24 (36%)	27 (28%)				5 (28%)	19 (39%)		
	HWE (p)		0.95	0.78				0.56	0.7		
**ACE2 (rs2285666)**											
Female	AA	101	1 (4%)	3 (4%)	0.83	1.00	23	0 (0%)	1 (6%)	0.18	0.29
	GA		5 (22%)	22 (28%)				0 (0%)	5 (31%)		
	GG		17 (74%)	53 (68%)				7 (100%)	10 (63%)		
Male	AA	63	13 (29%)	6 (33%)	0.97	1.00	45	3 (25%)	10 (30%)	1.00	1.00
	GG		32 (71%)	12 (67%)				9 (75%)	23 (70%)		
	HWE (p)		0.26	0.31				0.23	0.64		
**AGTR2 (rs1403543)**											
Female	GG	101	7 (30%)	22 (28%)	0.06	0.31	23	0 (0%)	7 (44%)	0.07	0.18
	AG		14 (61%)	32 (41%)				6 (86%)	8 (50%)		
	AA		2 (9%)	24 (31%)				1 (14%)	1 (6%)		
Male	GG	63	23 (51%)	9 (50%)	1.00	1.00	45	9 (75%)	14 (42%)	0.05	0.18
	AA		22 (49%)	9 (50%)				3 (25%)	19 (58%)		
	HWE (p)		0.18	0.11				0.58	0.67		

^a^ *p*-values were adjusted with the Benjamini–Hochberg method for multiple testing.

**Table 4 genes-13-01111-t004:** Association between the genotypes for ACE I/D, ACE2 rs2285666, and AGTR2 rs1403543 and retinal vessel diameters.

Polymorphism	Genotype	Veins Diameter		Arteries Diameter	
		Estimate (95% CI)	*p*	Estimate (95% CI)	*p*
ACE ^a^	II	1.96 (−9.18–13.09)	0.73	1.75 (−7.57–11.08)	0.71
	ID	−0.64 (−8.38–7.10)	0.87	0.89 (−5.80–7.58)	0.79
	DD	Ref.		Ref.	
** *Female* ** ^b^					
ACE2	AA	−9.6 (−44.05–24.85)	0.58	−3.11 (−26.88–20.65)	0.80
	GA	14.21 (−14.29–42.71)	0.32	10.94 (−10.52–32.41)	0.31
	GG	Ref.		Ref.	
AGTR2	AA	−2.43 (−26.30–21.44)	0.84	13.16 (−4.54–30.86)	0.14
	AG	7.85 (−8.41–24.11)	0.34	11.03 (−1.21–23.28)	0.08
	GG	Ref.		Ref.	
** *Male* ** ^b^					
ACE2	AA	1.14 (−8.87–11.15)	0.82	−0.67 (−9.43–8.09)	0.88
	GG	Ref.		Ref.	
AGTR2	AA	−3.55 (−11.99–4.88)	0.41	−6.41 (−13.69–0.87)	0.08
	GG	Ref.		Ref.	

^a^ Analysis adjusted to severity, sex, age, oxygen, LDH, and ferritin; ^b^ Analysis adjusted to severity, age, oxygen, LDH, and ferritin.

## Data Availability

Data can be provided upon request.

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
