# Peer review of "The Role of ACE, ACE2, and AGTR2 Polymorphisms in COVID-19 Severity and the Presence of COVID-19-Related Retinopathy"

_genes, 2022, doi:10.3390/genes13071111_

Round 1

Reviewer 1 Report

Authors have performed comparative study with the aim to determine the role of selected polymorphisms of genes in the RAAS pathway in the COVID-19 severity and their association with the presence of COVID-19 retinopathy. The authors concluded that  there is  a borderline association of the AGTR2 rs1403543-AA-genotype with COVID-19 retinopathy in males, hence the AGTR2 rs 1403543 A allele might represent a genetic risk factor for COVID-19 retinopathy in males.
This is an interesting study considering the state of the art. However, this reviewer has some questions and concerns to comment:

- The authors have used only vascular sings in their interpretation of COVID-19 related retinopathy. They should include more retinal findings to characterize this retinopathy, not only vascular alterations. If not, it would be convenience to change the definition to another one (i.e. COVID-19-related retinal vasculopathy). Please, see the references 10 and 11.

- Had these patients got other retinal alterations? As it is mentioned in methods section, the authors took eye fundus pictures using an OCT equipment, so it could be possible to observe structural alterations in the retina too. Please, clarify.

- Did these patients experience vision loss? It would be interesting to include this information in the tables.

- Why are there only significant correlations in the case of men?  What is the hypothesis or explanation for this? The authors should better explain this gender differences. It is poorly describe in the discussion section.

Author Response

We would like to thank the reviewer for their insightful comments. We have revised the manuscript accordingly.

- The authors have used only vascular sings in their interpretation of COVID-19 related retinopathy. They should include more retinal findings to characterize this retinopathy, not only vascular alterations. If not, it would be convenience to change the definition to another one (i.e. COVID-19-related retinal vasculopathy). Please, see the references 10 and 11.

COVID-19 retinopathy has been previously defined as the presence of either retinal hemorrhages, cotton wool spots, or vascular alterations, therefore, we have decided to use the same nomenclature. The presence of COVID-19 retinopathy in this study was defined as the presence of the retinal findings, namely retinal hemorrhages, cotton wool spots, or vascular alterations as is described in the Methods section (2.4. Image Analysis): »three researchers (KJ, AM and PJM) independently reviewed the fundus photographs for the presence of retinal findings, namely: retinal hemorrhages, cotton wool spots, dilated and tortuous vessels. Patients exhibiting any of the aforementioned findings were classified as having COVID-19 retinopathy«.

Even though the vascular alterations were more pronounced, other findings, including the cotton wool spots and retinal hemorrhages, have also been observed and are presented in Figure 1.

- Had these patients got other retinal alterations? As mentioned in the methods section, the authors took eye fundus pictures using OCT equipment, so it could be possible to observe structural alterations in the retina. Please, clarify.

The primary outcome of this study was the association of selected gene polymorphisms with the presence of retinal signs defined as COVID-19 retinopathy seen on fundus photographs. Therefore, we have used the color fundus photography setting of the DRI OCT Triton machine to obtain and analyze the color fundus photographs.

- Did these patients experience vision loss? It would be interesting to include this information in the tables.

None of the patients experienced vision loss. We have updated the results section: »None of the COVID-19 patients reported ocular symptoms such as itching, photophobia, foreign body, conjunctivitis, decreased visual acuity or vision loss.”

- Why are there only significant correlations in the case of men?  What is the hypothesis or explanation for this? The authors should better explain this gender differences. It is poorly describe in the discussion section.

We would like to thank the reviewer for this important request. We have further elaborated the possible explanations and rewritten the discussion as follows:

“The hypothesis of the protective role of the G-allele is further supported by the previous studies that have shown an association between the A-allele and increased risk of cardiovascular disease - namely ventricular hypertrophy in men and coronary ischemia in women 24.  One possible explanation of the gender differences that are also present in our study is that the AGTR2 gene is located on the X-chromosome, making males hemizygous, which affects gene expression. However, the male gender has been established as a risk factor for a severe course of COVID-19, possibly reflecting the gender-based differences in immune responses to COVID-19  32. Males were shown to have increased levels of pro-inflammatory chemokines and cytokines,whereas a more robust T-cell response was reported in females 32.  Therefore, our results could reflect the gender-based differences in the immunological response to SARS-CoV2 infection. Another possible explanation for our results is that they reflect a relatively small sample of females, as the p-value reaches significance when increasing the sample size.”

Reviewer 2 Report

The authors evaluate the correlation of polymorphisms in ACE 102 (rs4646994), ACE2 (rs2285666), AGTR2 (rs1403543) with the presence of COVID-19, the severity of COVID-19, and the presence of COVID-19 retinopathy. They found borderline association of the AGTR2 rs1403543 AA genotype with increased risk of COVID-19 retinopathy in males. This article is well-organized, and the discussion is insightful.

1.   What is the severity of the COVID-19 (like respiratory system) in these patients with COVID-19 retinopathy? Did the authors find an association between the severity of the COVID-19 and the incidence/severity of COVID-19 retinopathy?

Author Response

We would like to thank the reviewer for pointing out this important question. COVID-19 retinopathy was observed in both groups but was more pronounced in the severe group. We have initially included more patients in the study, but then had to exclude the patients without a successful gene analysis. When we performed statistical analysis of association between COVID-19 severity on the presence of COVID-19-related retinopathy with the initial data set, the severity was shown to affect the presence of retinopathy and patients with severe disease were shown to have a 4- fold increased risk of retinopathy (p= 0,006). However, after excluding patients without a successful genetic analysis (n=6), the significance is borderline (p=0,05).

We thank you again for your overall positive review.

Round 2

Reviewer 1 Report

The authors have corrected their manuscript taking into account my suggestions and remarks. I now think that the paper is acceptable.